# Photoluminescence imaging of single photon emitters within nanoscale strain profiles in monolayer WSe$_2$

Artem N. Abramov[1], Igor Y. Chestnov[1], Ekaterina S. Alimova[2], Tatiana Ivanova[1], Ivan S. Mukhin[1,3], Dmitry N. Krizhanovskii [4], Ivan A. Shelykh[1,5,6,7], Ivan V. Iorsh [1,6,7] & Vasily Kravtsov[1] ✉

Local deformation of atomically thin van der Waals materials provides a powerful approach to create site-controlled chip-compatible single-photon emitters (SPEs). However, the microscopic mechanisms underlying the formation of such strain-induced SPEs are still not fully clear, which hinders further efforts in their deterministic integration with nanophotonic structures for developing practical on-chip sources of quantum light. Here we investigate SPEs with single-photon purity up to 98% created in monolayer WSe$_2$ via nanoindentation. Using photoluminescence imaging in combination with atomic force microscopy, we locate single-photon emitting sites on a deep sub-wavelength spatial scale and reconstruct the details of the surrounding local strain potential. The obtained results suggest that the origin of the observed single-photon emission is likely related to strain-induced spectral shift of dark excitonic states and their hybridization with localized states of individual defects.

Single-photon emitters (SPEs) are key elements of rapidly developing quantum technologies as they produce individual photons - basic carriers of quantum information[1,2]. One of the very promising material platforms for creating SPEs is the family of two-dimensional (2D) van der Waals crystals[3]. Among them, monolayers of transition metal dichalcogenides (TMDs) hold a special place as direct bandgap semiconductors[4] with large exciton binding energies[5] that can host localized SPEs characterized by high brightness, narrow spectral lines, and decent single-photon purity[6–10].

Due to their 2D nature and unlike traditional bulk semiconductors, TMD monolayers can be easily locally strained, for example, via deformation in the out-of-plane direction[11,12]. Since the associated excitonic properties are highly strain-dependent, this results in effective local potentials for excitons, providing a powerful approach[13] towards creating, activating, or tuning SPEs in TMD monolayers with nanostructured substrates and

nanoparticles, piezoelectric devices, or atomic force microscope (AFM) probes[14–18].

In order to enable practical applications of SPEs, their integration with on-chip nanophotonic resonators and waveguides is highly desirable. As previously demonstrated for solid-state SPEs, coupling to optical structures can significantly improve the emission rate, reduce radiative lifetimes, and increase indistinguishability of emitted photons, as well as facilitate their routing into required optical channels via Purcell effect[19,20]. Similarly, integrating strain-induced SPEs in TMD monolayers with nanophotonic structures allows optimization and control of their emission properties such as directivity and quantum yield as has been demonstrated recently[21,22].

However, it is challenging to achieve optimal coupling since SPEs must be placed at the maximum of local optical field distribution. This requires accurate knowledge of the actual SPE position on the nanoscale within a larger strained region of TMD monolayer, which is

[1]School of Physics and Engineering, ITMO University, Saint Petersburg 197101, Russia. [2]Peter The Great St. Petersburg Polytechnic University, Saint Petersburg 195251, Russia. [3]St. Petersburg Academic University, Saint Petersburg 194021, Russia. [4]Department of Physics and Astronomy, University of Sheffield, Sheffield S3 7RH, UK. [5]Science Institute, University of Iceland, Dunhagi-3, IS-107 Reykjavik, Iceland. [6]Abrikosov Center for Theoretical Physics, MIPT, Dolgoprudnyi, Moscow Region, 141701, Russia. [7]Russian Quantum Center, Skolkovo, Moscow 143025, Russia. ✉e-mail: vasily.kravtsov@metalab.ifmo.ru

not readily available. The issue is further complicated by the fact that the exact mechanisms underlying the formation of strain-induced SPEs in monolayer semiconductors are still not fully clarified. In particular, questions remain about the roles of atomic defects and strain gradients for the SPE formation process[23], as well as possible contributions from bright/dark excitons and hybridization effects[24].

Here we experimentally investigate nanoscale positioning and polarization properties of individual SPEs created in monolayer WSe$_2$ via nanoindentation. The SPEs exhibit excellent single-photon purity with second order correlation function values down to $g^{(2)}(0) \sim 0.02$ and high emission rates of $> 1$ MHz, which allows us to locate SPEs on a deep subwavelength spatial scale through photoluminescence (PL) imaging. By reconstructing strain distribution maps from the measured AFM topography, we extract local strain tensor components at the exact SPE location and in its vicinity. The observed correlation between the experimentally measured emission properties and theoretically calculated local deformation parameters allows us to conclude on the likely mechanism of SPE formation, which relies on hybridization between dark excitons and localized defect states. Our work establishes a general method for investigating properties of quantum emitters formed in complex 2D nanoscale strain potentials and offers new insights for the future development of practical SPEs based on strained 2D semiconductors for applications in quantum communications and computing.

## Results

### SPEs with high single-photon purity and brightness

In our experiment, we create strain-induced SPEs in monolayer WSe$_2$ via nanoindentation with an AFM probe as shown in Fig. 1a, following the general approach demonstrated in Ref. 17. To be able to accurately locate SPE sites on a subwavelength scale, we first fabricate cross-

shaped Ag alignment marks with 1 $\mu$m width and 60 nm height on a SiO$_2$(1 $\mu$m)/Si substrate. A thin layer of polymethyl methacrylate (PMMA) is then spun onto the substrate, and a large-scale high-quality WSe$_2$ monolayer is transferred on top to overlap with the alignment marks as shown in the optical microscope image in Fig. 1b. We create an array pattern of nanoindented regions in the monolayer using a specially modified Si AFM probe (see Supplementary Note 1), where plastic deformation in the PMMA layer in combination with adhesion at the PMMA/WSe$_2$ interface result in local straining of the monolayer.

For optical measurements, the sample is mounted in a cryostat and maintained at a temperature of 7 K (for setup details, see Methods and Supplementary Note 2). Under focused excitation by a continuous wave HeNe laser with 633 nm wavelength, locally deformed WSe$_2$ monolayer regions emit PL spectra consisting of bright and narrow spectral lines redshifted with respect to the free neutral exciton peak. Fig. 1c demonstrates an example PL spectrum collected from a selected strained region, where the peak at 710.5 nm is attributed to the free neutral exciton $X_b^0$, while peak at 764.4 nm is attributed to a single photon emitter. We observe similar individual spectral lines on more than 50% of all indented regions in the monolayer, with the distribution of their wavelengths covering the range of 720 – 800 nm and peaked at 750 nm as described later in the text. For a typical emitter, we measure degree of linear polarization of more than 70% (an example of polarization-resolved PL signal is shown in the inset of Fig. 1c), which allows us to maximize the SPE emission intensity with respect to the broader mostly unpolarized PL background (see Supplementary Note 3).

The individual spectral lines in the PL spectra exhibit a saturating dependence on excitation power, which is characteristic of emission from a single two-level system. From the experimentally measured power dependencies of PL intensity, we determine the saturation

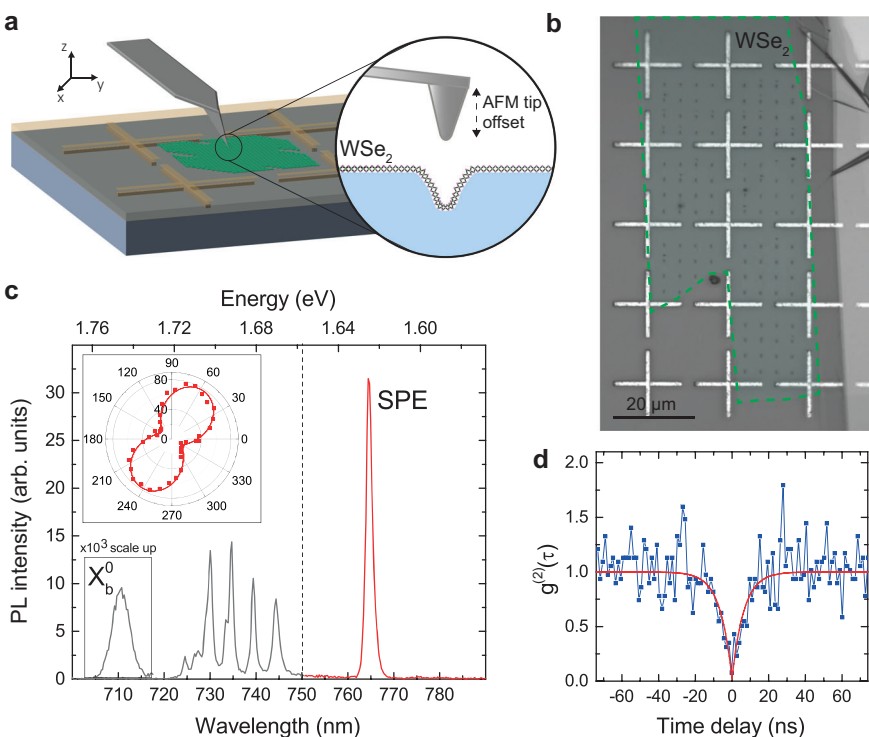

**Fig. 1 | Single-photon emission in a locally strained WSe$_2$ monolayer sample with alignment marks. a** Schematic of single-photon emitter (SPE) fabrication by nanoindentation in a WSe$_2$ monolayer transferred onto a substrate with an alignment pattern. **b** Optical image of the sample showing arrays of alignment marks and nanoindents. The WSe$_2$ monolayer border is highlighted by green dashed line. **c** Photoluminescence (PL) spectrum from a selected SPE. The spectral part

detected in the experiment is highlighted with red color. Spectral peak of the free bright neutral exciton $X_b^0$ at $\lambda = 710.5$ nm is scaled up by $10^3$. The inset shows typical polarization-resolved spectrally integrated PL signal from an SPE. **d** Second-order correlation function $g^{(2)}(\tau)$ measured for the SPE spectral peak (blue dots) together with fit (red curve), which demonstrates antibunching.

power and corresponding emission rate as outlined in detail in Supplementary Note 4. For most of our measurements, the excitation laser power is chosen slightly below the saturation power and lies in a 0.7–1.0 $\mu$W range, which corresponds to emission rates of ~1 MHz limited by relatively low PL quantum yield of 1–4%. We verify the single-photon character of the emitted PL by measuring the second-order autocorrelation $g^{(2)}(\tau)$ as a function of time delay $\tau$ in a Hanbury Brown and Twiss setup and fitting the data with $g^{(2)}(\tau) = 1 - (1 - A)e^{-|\tau/\tau_0|}$, where $A$ and $\tau_0$ are fit parameters, and there are no corrections for background or dark counts. The extracted values of $\tau_0$ are roughly of the same order as the SPE PL decay times lying in a 1–20 ns range. All extracted values of $A = g^{(2)}(0)$ are below 0.4, and a significant portion of them are below 0.1 for individual redshifted PL peaks from different indented regions, confirming the single-photon nature of the corresponding localized emitters. While the average value of $g^{(2)}(0)$ in our statistics is $0.15 \pm 0.12$ (see Supplementary Note 5), some emitters exhibit significantly smaller values down to $g^{(2)}(0) \simeq 0.02$ as demonstrated in Fig. 1d, which shows experimental $g^{(2)}(\tau)$ data (blue dots) together with a corresponding fit (red line). We note that the estimated single-photon purity of 98% is among the highest reported for SPEs in atomically thin semiconductors to date.

## SPE coordinates within nanoscale strain profiles

While each fabricated locally strained region in monolayer WSe$_2$ extends over 0.5 $\mu$m, the SPEs are expected to be point-like and localized at well-defined sites within the strain profiles. To accurately locate the SPEs, we use the PL-imaging approach that has been previously applied for single molecules and individual solid-state semiconductor quantum dots[25,26]. The approach is based on fitting the real-space PL image of the SPE with a point spread function to determine the SPE coordinates with respect to alignment marks. We excite individual strained regions and select corresponding SPE PL signals via spectral (wavelength > 750 nm, red color in Fig. 1c) and polarization filtering. The sample is simultaneously illuminated by white light from a lamp spectrally filtered with a 10 nm band pass filter to detect reflected optical signal from the Ag alignment marks, and both PL and reflected images of the sample are captured by a CMOS camera.

Fig. 2a shows a selected camera image, where the bright circle corresponds to the PL signal from a strain-induced SPE in monolayer WSe$_2$. To determine the coordinates of the SPE and alignment marks, we take line cuts from the camera image along horizontal ($X$) and vertical ($Y$) directions, with the background-subtracted data shown in Fig. 2b and c (dots), respectively. We note that the relative intensities of the peaks shown in Fig. 2b, c are controlled by the intensity of the alignment mark illumination and chosen to maximize corresponding signal-to-noise ratios and minimize the uncertainty in the coordinate extraction procedure. While the spatial distribution of PL from a single point emitter in an ideal visualization system follows the Airy function, we use Gauss functions as a good approximation for their central parts, which is more convenient for data processing[25,27]. From Gaussian fits of the marks (blue curves) and SPE (red curves) profiles, we extract the $X$ and $Y$ coordinates as peak centers together with corresponding uncertainties as fit errors. The uncertainties of the $X(Y)$ coordinates for the fits shown in Fig. 2b and c are obtained as 5.5 nm (6.6 nm) for the SPE and 7.9 nm (13.3 nm) for the alignment mark.

Next, we map the extracted coordinates onto the AFM topography image of the sample shown in Fig. 2d. We determine the coordinates of the alignment marks in the AFM image, match them to the corresponding coordinates in the optical image, obtain the associated transformation matrix, and use it to locate the SPE site on the AFM map. We note that despite the presence of the PMMA layer, the alignment marks create enough protrusion at the surface to allow extracting their center $X$ and $Y$ coordinates with accuracy of 1.6 nm and 3.4 nm, respectively, as shown in Fig. 2e and f. Based on the uncertainties obtained for SPE and alignment marks coordinates in the

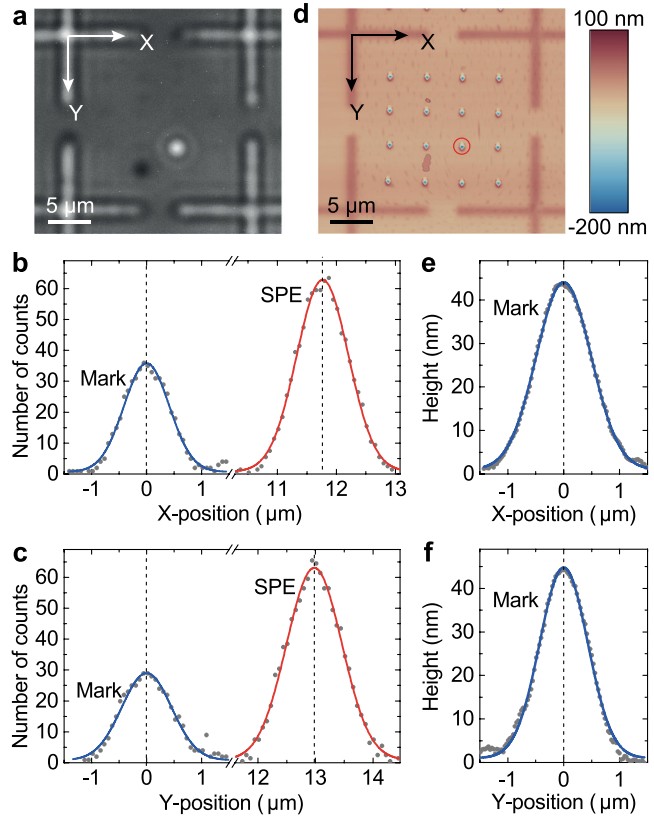

**Fig. 2 | Measurement of SPE position with subwavelength accuracy. a** Optical image acquired by illuminating the sample simultaneously by a HeNe laser and 10 nm band pass filtered light from a halogen lamp. The circular spot corresponds to PL from an SPE. **b, c** Selected X- and Y-line cuts of the camera image taken through the alignment mark and SPE (dots), with respective Gaussian fits (blue and red curves). Signals are not normalized. **d** AFM topography image of the same region, with the red circle indicating the laser-illuminated nanoindent. Color scale represents the height. **e, f** Selected X- and Y-line cuts of the AFM scan taken through the alignment mark, with Gaussian fits (blue curves). Vertical dashed lines indicate the extracted coordinates.

optical and AFM images we estimate the expected uncertainty of the SPE position on the AFM map as ~22(28) nm for the $X(Y)$ coordinates. Further, we take a sequence of PL images and calculate the experimental uncertainty as the standard deviation of the extracted SPE position, which gives ~30(13) nm for the $X(Y)$ coordinates of the SPE shown in Fig. 2, consistent with the estimated values. The extracted uncertainties are likely limited by the residual vibrations and drift in our cryostat system and can be reduced down to 10-20 nm in a setup with improved stability. We note that an additional uncertainty can in principle be present due to the different sample temperatures for the optical and AFM measurements and the associated thermal expansion. However, for the materials and characteristic alignment grid sizes used in our experiment the potential distortions are estimated to be below 10 nm.

The AFM scan of a single nanoindent along with the identified SPE location are shown in Fig. 3a. Surprisingly, the SPE appears at the outer slope of the fold wrapped around the indentation area rather than in the center of the nanoindent. Similar results are observed for other studied SPEs in our sample (see Supplementary Note 6 and Fig. 4). These observations do not align well with the common interpretation of the SPE origin in TMD monolayers relying on strain-induced nanoscale trapping of excitons[14,15,28–30]. While exciton trapping is expected at the energy minima of the strain-induced potential, that is, at the maxima of the local tensile strain, the SPEs in our experiment are

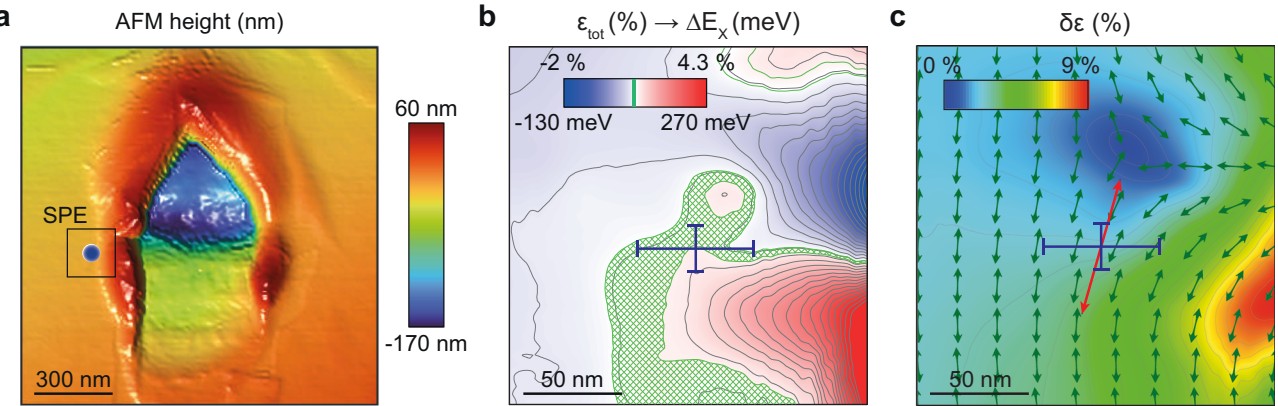

**Fig. 3 | Analysis of local strain in the vicinity of the SPE site. a** AFM topography of the strained region with the experimentally extracted SPE location indicated with a blue circle. **b** Calculated from the AFM topography total in-plane strain $\epsilon_{tot}$ (false color scale in %), which translates into exciton transition energy shift $\Delta E_X$ (scale in meV). The region shown corresponds to the area indicated with a black rectangle in (**a**), and the SPE location together with experimental uncertainties obtained from a series of measurements are shown with the blue error bars. In the green-shaded region, the calculated strain-shifted dark exciton energy matches the SPE energy. **c** Calculated spatial distribution of the local strain difference along the principal axes $\delta\epsilon$ (false color image) quantifying the strain asymmetry, together with the calculated orientation of the principal strain axis corresponding to the lowest-energy state of the exciton radiative doublet (green arrows). The SPE location and polarization direction are indicated with blue error bars and long red arrow, respectively.

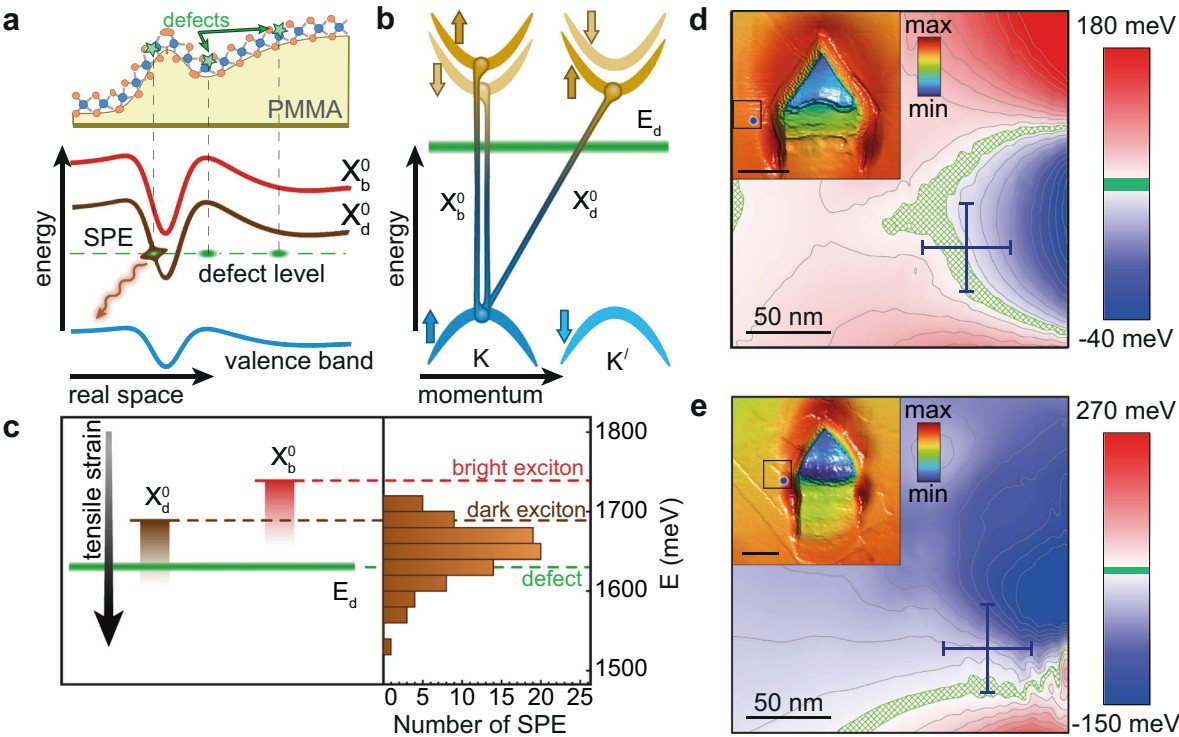

**Fig. 4 | Single-photon emission mechanism in strained monolayer WSe₂.** **a** Spatial variation of the exciton energy due to nonuniform strain distribution in the deformed monolayer. Both bright $X_b^0$ and dark $X_d^0$ exciton bands are redshifted in the region of local tensile strain. The SPE originates from hybridization of the dark exciton with the mid-gap defect state (indicated by green circles) when they come in resonance (the defect energy is indicated by the horizontal green dashed line, and photon emission process is indicated by the wavy red arrow). **b** Schematic illustration of the dark intervalley exciton formed in monolayer WSe₂ between the $K$ and $K'$ valleys, with arrows indicating spin states. Mid-gap defect level is shown with green line. **c** Right panel: measured distribution of SPE energies for a large ensemble of emitters. Left panel: schematic illustration for the spectral positions of the free bright $X_b^0$, dark $X_d^0$, and the localized defect $E_d$ exciton states shown by dashed horizontal lines, with the expected effect of the strain-induced redshift (indicated by red and brown gradient shaded areas). **d, e** Maps of the total in-plane strain $\epsilon_{tot}$ near 2 additional SPE sites, with AFM topography images in the inset (AFM scale bar is 300 nm, color scale represents the height). In the green-shaded regions, the calculated strain-shifted dark exciton energy matches the SPE energy. The experimental uncertainties for extracted SPE coordinates obtained from a series of measurements are indicated with blue error bars.

located far from the strain maxima. We note that a similar trend has been recently observed for strain-induced SPEs in WS$_2$[31].

## Calculation of the local strain tensor components

To understand the possible origin of the studied SPEs, we analyze the local strain distribution in the nanoindent vicinity. In 2D crystals, the strain can be described by the in-plane tensor $\hat{\epsilon}$[32]:

$$\epsilon_{ij} = \frac{1}{2}\left(\partial_i u_j + \partial_j u_i + \partial_i h \partial_j h\right). \qquad (1)$$

Here $i, j = x, y$, and $\boldsymbol{u}(\boldsymbol{r}) = (u_x(\boldsymbol{r}), u_y(\boldsymbol{r}))^\top$ stands for the in-plane displacement vector, while $h(\boldsymbol{r})$ is the out-of plane deformation profile known from the AFM data. In the absence of external in-plane forces, the $\boldsymbol{u}(\boldsymbol{r})$-dependence can be retrieved from the equilibrium condition $\boldsymbol{\nabla} \cdot \hat{\sigma} = 0$ and the Hooke's law

$$\begin{pmatrix} \sigma_{xx} \\ \sigma_{yy} \end{pmatrix} = \frac{E}{1-\nu^2}\begin{pmatrix} 1 & \nu \\ \nu & 1 \end{pmatrix}\begin{pmatrix} \epsilon_{xx} \\ \epsilon_{yy} \end{pmatrix}, \quad \sigma_{xy} = \frac{E}{1+\nu}\epsilon_{xy}. \qquad (2)$$

Here $\hat{\sigma}$ is the stress tensor and $\boldsymbol{\nabla} = \left(\partial_x, \partial_y\right)$ is a divergence row-vector, $E$ is the 2D Young's modulus and $\nu$ is the Poisson's ratio[33,34]. From combined Eqs. (1) and (2), we obtain an equation for the displacement vector $\boldsymbol{u}(\boldsymbol{r})$, solve it, and finally calculate the full strain tensor $\hat{\epsilon}(\boldsymbol{r})$ as a function of the in-plane coordinate $\boldsymbol{r}$ according to Eq. (1) (for more detailed description of the calculation procedure, see Methods and Supplementary Note 7).

From the calculated strain tensor, we extract spatial distributions in the vicinity of the SPE site for the following three parameters: (i) total in-plane strain $\epsilon_{\text{tot}} = \epsilon_{xx} + \epsilon_{yy}$ plotted in Fig. 3b, (ii) orientation $\theta$ of the principal strain axis plotted in Fig. 3c with green arrows, and (iii) strain difference along the principal axes $\delta\epsilon$ plotted in Fig. 3c as a false-color map. The region shown in Fig. 3b, and c is a zoomed-in area indicated with a black rectangle in Fig. 3a. The experimentally extracted SPE location is indicated with a blue circle in Fig. 3a and with blue error bars in Fig. 3b and c.

## Discussion

We first discuss the results for parameters $\theta$ and $\delta\epsilon$, which are associated with the anisotropy of local strain fields (see Methods for calculation details). In monolayer TMDs, anisotropic strain breaks the chiral selection rules and results in a radiative excitonic doublet with emission linearly polarized in two orthogonal directions defined by the principal strain axes[7,35,36]. Consistent with this theoretical expectation, our experimentally measured linear polarization direction for the SPE indicated in Fig. 3c with a red arrow is found to coincide with the principal strain axis (green arrows) within the spatial area defined by the experimental uncertainty for the SPE location (blue error bars for $X$ and $Y$ uncertainties). On the other hand, our SPE spectral peaks do not exhibit fine-structure splitting and emit photons in a single dominant linear polarization. This is similar to results reported previously for structures with strong asymmetry such as nanowrinkles or artificial quasi-1D confinement potentials where PL is linearly polarized along the elongated direction[37–40]. We estimate the expected fine-structure splitting of the SPE spectral peak from local strain anisotropy governed by $\delta\epsilon$. As observed in Fig. 3c, the strain difference $\delta\epsilon$ at the SPE location is ~2.4%, which translates to ~2.4 meV strain-induced contribution to the exciton fine-structure splitting if we use the estimate of 1 meV/% from Ref. 35. This is similar to the linewidth of our emitters of 2–3 meV (see Supplementary Note 5), which makes it challenging to distinguish the split peaks in the measurements. Additionally, at the temperature of 7 K, and taking into account relatively long measured lifetimes on the ns scale, the lower-energy level in the doublet will be populated predominantly, which might also explain the absence of SPE peak splitting in our experimentally obtained data. We note that this is consistent with the experimental observation that the SPE polarization matches the principal axis corresponding to the lowest-energy state (green arrows in Fig. 3c) in the radiative doublet.

Next, we discuss the results for the total strain $\epsilon_{\text{tot}}$ with calculated spatial distribution in the vicinity of the SPE site shown in Fig. 3b (false color scale in %). Assuming that the energy bandgap depends linearly on the total in-plane strain and that the exciton binding energy is affected only weakly, we calculate the expected shift of the exciton transition energy as $\Delta E_X = \alpha \epsilon_{\text{tot}}$, where $\alpha = 63$ meV/% is the estimate for the energy shift rate for monolayer WSe$_2$ taken from Ref. 34. As observed in Fig. 3b, the spatial distribution of the strain-induced energy shift (false color scale in meV) does not exhibit relevant conditions for exciton trapping. The quantum confinement of excitons can be expected to emerge under the strain modulation on the order of 1% within 10 nm length scale[41], while the extracted energy landscape appears to be smooth on a significantly larger spatial scale (less than 0.05% within 10 nm). In addition, the strain-free energy of $E \sim 1.703$ eV estimated from the redshift due to deformation (~78 meV) at position of the SPE with wavelength of 763 nm (1.625 eV) turns out to be far from the strain-free bright exciton energy $E_{Xb} = 1.745$ eV.

These observations suggest that strain-induced trapping of bright excitons is an unlikely SPE formation mechanism in our experiment. Instead, they are consistent with the hypothesis that attributes SPE formation to the strain-induced resonant hybridization of optically dark excitons with point-like defects as discussed in recent works[23,24,42]. The proposed single-photon emission mechanism is schematically illustrated in Fig. 4a and relies on brightening of optically dark excitons ($X_d^0$) when scattering on the individual point-like defects (green dots) present in the WSe$_2$ monolayer. To enable the brightening process, the dark exciton ($X_d^0$, brown curve) should come in resonance with the defect level (green dashed line), which is achieved via strain-induced energy shift. While the energy of a defect level is almost independent of strain[42], the dark exciton energy follows the local variation of the bandgap and matches the defect energy at certain values of total in-plane strain resulting in a possible photon emission event (wavy arrow).

We note that the dark excitons involved in this mechanism are likely *intervalley* excitons composed of the same-spin valence and conduction band states from the neighboring valleys, see Fig. 4b. In WSe$_2$ the dark intervalley exciton $X_d^0$ lies ~45 meV below the bright exciton $X_b^0$[43,44] and almost coincides in energy with the spin-forbidden intravalley dark exciton[33,34,45]. However, brightening of intravalley dark excitons is less likely as it requires an additional spin flip process. The defects involved in the brightening process are likely the omnipresent Se vacancies[23], with corresponding energy levels lying several tens of meV below the dark exciton in monolayer WSe$_2$[24,42]. Our experimentally measured distribution of SPE energies obtained from a large ensemble of >80 emitters shown in Fig. 4c (right panel) supports this assignment as it is peaked near 1.63 eV corresponding to the expected spectral position of the mid-gap Se-vacancy defect[42]. The relatively large width of the SPE energy distribution agrees with the results of recent experimental studies on statistics of emitters in WSe$_2$ monolayer[46,47] and reports on dark-localized exciton mixing in the wide spectral range around the defect energy level[42].

As schematically illustrated in the left panel of Fig. 4c, the local strain at the SPE site brings the energy of the dark exciton ($X_d^0$) in resonance with the defect level. This is indeed what is observed in our data shown in Fig. 3b. There we highlight with green color the spatial region where the strain-induced energy redshift falls within the range 68-78 meV, corresponding to the difference between the energies of the given SPE (1.625 eV) and dark intervalley exciton (~1.70 eV) in monolayer WSe$_2$. Analogous data extracted for two other SPEs are shown in Fig. 4d (64–74 meV redshift) and e (53–63 meV redshift), together with corresponding AFM images with indicated extracted SPE locations (insets). All three plots demonstrate that SPEs are located at

or very near the corresponding green-highlighted regions where the difference between the SPE and dark exciton energies is equal to the strain-induced energy redshift. The data for the set of emitters are summarized in Supplementary Note 8. Analysis of the SPE brightness demonstrates that the emission intensity is maximized near 1.63 eV where the SPE energy matches the expected Se-vacancy energy level[42]. These observations, together with the lack of evidence for bright exciton trapping in our experimental data and theoretical analysis, render the resonant hybridization of dark excitons with individual atomic defects a favorable explanation of SPE formation in our strained samples of monolayer $WSe_2$.

In summary, we have investigated the precise nanoscale distribution of mechanical strain in the vicinity of single-photon emitting sites fabricated via nanoindentation in monolayer $WSe_2$ using combined PL and AFM imaging. The high single-photon purity and brightness of the fabricated SPEs allowed us to locate them within the strained region with subwavelength accuracy on the order of tens of nm. To obtain insights into the SPE formation mechanism, we provide a model for calculating the full strain tensor at and near the determined SPE location. Our analysis and comparison of the experimentally measured data and extracted components of the local strain tensor shows that the single-photon emission in our case likely relies on brightening of dark excitonic states on individual point-like defects in the monolayer, rather than on formation of quantized excitonic levels due to exciton trapping. The obtained results provide important insights into the microscopic mechanisms governing single-photon emission in 2D semiconductors and contribute towards further integration of SPEs in TMD monolayers with nanophotonic resonators and waveguides.

## Methods

### Sample preparation
Ag alignment marks are fabricated on a $SiO_2(1\,\mu m)$/Si substrate using electron-beam lithography with double PMMA layer followed by thermal evaporation of 2 nm/60 nm Cr/Ag and lift-off in ultrasound with acetone. The marks form a grid of $4 \times 4$ squares of $20 \times 20\,\mu m^2$ each and are covered with a 300 nm PMMA layer. $WSe_2$ monolayers are obtained from a bulk crystal by mechanical exfoliation and transferred onto the PMMA surface using standard dry transfer. Nanoindents in $WSe_2$ monolayer with depth of 100–200 nm are formed using silicon AFM probe blunted with a focused ion beam.

### PL-imaging approach
To obtain a PL image, the sample is illuminated with HeNe laser to excite photoluminescence and lamp light with a bandpass filter to illuminate alignment marks. The signal is collected by a $50 \times$ long working distance Mitutoyo objective (NA = 0.65) mounted on a three-axis piezoelectric translator for the precise positioning of the laser spot. Data acquisition time and lamp brightness are optimized to obtain similar number of counts for the SPE PL and reflection signal from the alignment marks, as well as to minimize the reflection of the lamp light from the substrate and minimize possible sample drift during measurement.

### Strain analysis
The strain profile is obtained from the solution of the equilibrium equation $\nabla \cdot \hat{\sigma} = 0$ for the in-plane displacement vector $\boldsymbol{u}$ using finite element method with $\nu = 0.196$[33,34]. Orientation of the principal strain axes is obtained by diagonalizing the strain tensor at every point of the AFM scan grid. This requires rotation at the angle $\theta$ given by $\tan(2\theta) = 2\epsilon_{xy}/(\epsilon_{xx} - \epsilon_{yy})$ which eliminates shear strain and brings $\hat{\epsilon}$ to the diagonal form. The difference of the principal strains $\delta\epsilon = \sqrt{(\epsilon_{xx} - \epsilon_{yy})^2 + 4\epsilon_{xy}^2}$ governs the fine-structure splitting with the rate $\Delta_{\mathrm{fs}}$ whose value is assumed to be below 1 meV per percent of strain according to the estimations for the bright exciton doublet in $WSe_2$[35]. The vector map in Fig. 3c corresponds to the principal strain of the

state defined by the strain tensor eigenvector $(-\sin(\theta), \cos(\theta))^\top$ which corresponds to the lowest in energy state of the fine-structure doublet. This state is expected to dominate polarization properties of the SPE.

## Data availability
The data that support the findings of this study are available within the main text and Supplementary Information. Any other relevant data are available from the corresponding authors upon request.

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

## Acknowledgements

This research was supported by Priority 2030 Federal Academic Leadership Program. The authors acknowledge funding from Ministry of Science and Higher Education of Russian Federation, goszadanie no. 2019-1246. The work of I.V.I. was supported by Rosatom in the framework of the Roadmap for Quantum computing (Contract No. 868-1.3-15/15-2021 dated October 5). I.A.S. acknowledges support from the Icelandic Research Fund (Project "Hybrid polaritonics"). This work was carried out using equipment of the SPbU Resource Centers "Nanophotonics" and "Nanotechnology". We thank Mikhail Glazov and Ekaterina Khestanova for fruitful discussions.

## Author contributions

A.N.A., I.V.I. and V.K. conceived the idea and designed the experiment. A.N.A. and T.I. fabricated TMD monolayers. A.N.A. and I.S.M. fabricated nanoindentors and alignment grids. A.N.A. fabricated and characterized SPE arrays. A.N.A. and E.S.A. performed optical measurements. I.Y.C. and I.V.I. developed theoretical model and performed calculations. A.N.A., I.Y.C. and V.K. wrote the manuscript. A.N.A., I.Y.C., D.N.K., I.A.S., I.V.I. and V.K. discussed and interpreted the results. V.K. supervised the project.

## Competing interests

The authors declare no competing interests.
