## [Peer Review File · Nature Communications]

REVIEWER COMMENTS

Reviewer #1 (Remarks to the Author):

The manuscript by A. N. Abramov and colleagues addresses a timely topic concerning the physical origin of single photon emitters in two-dimensional materials. Many groups worldwide study this kind of sources aiming at their exploitation for quantum technologies.

This manuscript presents a series of cleverly conducted experiments. The paper is clearly written and the results well presented. Based on the reported experiments and related analyses the authors confirm and strengthen a previously published hypothesis concerning the mechanism leading to the generation of single photons in 2D materials (Refs 23 and 45). The concomitant presence of moderate strain and of defects triggers the optical emission from otherwise dark excitons. To this respect, the present work does not bring great elements of novelty.

In the following, a few technical concerns are listed.

- 1) Are the emission spectra reproducible in terms of intensity and emitted photon wavelength after each thermal cycle?
- 2) The relationship between saturation power and emission rate might be not known to all readers. Please, provide a brief explanation; the Supporting Information could be an appropriate place.
- 3) If feasible, the authors should provide a statistical analysis regarding of characteristics of the observed single photon emitters. In addition to the emission wavelength histogram in Fig. 4 (c), it'd be relevant to present a similar analysis on the single photon emitter linewidth and $g^2(0)$, if available.
- 4) The matching between the AFM and the optical microscopy measurements is made from images recorded at different temperatures. This is of course fine. Nevertheless, can the authors comment on the reliability of this method? May issues related to the thermal expansion/contraction of the sample be present?
- 5) Frankly, I could not follow completely the procedure employed to extract the beautiful maps shown in Figures 3 (b) and (c) and in Figures 4 (d) and (e). In particular, I refer to equations (1) and (2) and to the method employed to derive the strain tensor over the interested area. The known quantities are the material's elastic parameters and the AFM height profile (namely, the displacement h). I suppose then that from the vertical displacement, via the material's elastic parameters, the authors retrieve the in-plane displacement vector (u). It looks like a reverse Poisson's effect: known the vertical displacement, we derive the in-plane compression/expansion of the layer. I believe that this important aspect would deserve a more detailed discussion in the Supporting Information.

6) The discussion at page 9 about the fine-structure splitting is a bit ill-posed given the relatively broad linewidth of the emitter shown in Figure 1 for example.

7) Figure 4 (c) is very suggestive. I wonder if the vertical axis of the left panel corresponds to a true energy scale. Namely, do the intervals spanned by the horizontal lines relative to X_{bright} and X_{dark} correspond to a specific strain value?

A curiosity: what do the authors expect to observe (or have observed) if X_{bright} is resonant with the defect level? Actually, this seems to be a real possibility given the large tensile strain values present in the imprinted monolayers (dark blue regions in Figures 3 (b) and (c) and in Figures 4 (d) and (e)).

In summary, this is a well written manuscript presenting a series of interesting experimental results and elegant numerical analysis (provided this latter is more clearly presented and detailed). The main conclusions of this work match with those previously reported and therefore the novelty of this work is somehow undermined. My opinion is that npj 2D materials and applications could be a more suitable journal with the advantage of a rapid transfer and publication.

Reviewer #2 (Remarks to the Author):

The manuscript by Abramov et al. describes their study of strain induced single photon emitters in monolayer WSe₂ using a nanoindentation process. Through PL imaging and atomic force microscopy, they are then able to deduce the amount of strain at the vicinity of the emitter. This then allows them to calculate what the strain induced shift of the dark exciton in WSe₂ would be at the emitter location. From here, they show a high degree of correlation between the energy shifted dark exciton wavelength and the emitter wavelength.

While neither the nanoindentation process for generating single photon emitters nor the PL/AFM imaging for locating a quantum emitter are novel, I do feel that the overall conclusions of the paper are very noteworthy. The underlying mechanisms for strain induced single photon emission in monolayer WSe₂ has not been well understood, with one prominent theory being exciton funneling and band shifting at the strain locations. The work presented here shows that exciton funneling is not correct and that hybridization of dark exciton states with Se defects is the most probable cause. These results will be of much interest to those working in quantum emission from 2D materials and, as such, I feel the manuscript is worthy of consideration for publication in Nature Communications.

Below are a few technical comments:

1) In figure 1d, the fit definitely extends well below the histogram data point at $g_2(0)$, with the data point appearing to be roughly $g_2(0) = 0.1$. This fact, coupled with the level of noise in the $g_2(t)$ spectra, makes it hard for me to be fully convinced that the $g_2(0)$ value is 0.02 as claimed. Calculating and listing the uncertainty in the fit might be helpful.

2) In supplementary note 3, I think it might be illustrative to include an additional figure showing again the filtered spectra but this time with it normalized to the unfiltered SPE PL. Maybe it is just due to the normalization but to me it appears that you are able to reduce the background signal by more than 50% using polarization filtering which I do not believe should be possible. To my understanding, this background signal is usually unpolarized.

3) In supplementary note 5, the text references figure S3 when it should be figure S5. Please check the manuscript for other such small typos.

Reviewer #3 (Remarks to the Author):

Abramov and co-authors present measurements on single photon emitters in strained WSe₂. They provide spatially resolved AFM images aligned with optical PL images, to quite accurately pinpoint the location of the single photon emitters relative to local strain.

Single photon emitters are a key ingredient of many quantum technologies. The present paper confirms strain-induced hybridization between conduction band states and localized defect states as their origin in WSe₂. The paper is well written, the topic at hand is timely, and the contribution significant. I therefore recommend publication in Nature Communication.

Suggestions:

-> As far as I understand, the method by the authors would allow for changing the relative position between the strain-inducing AFM tip and the defect. Have the authors tried this, or is the number of defects too large and the membrane relaxation too big to find the same defect again?

-> Can the authors provide quantitative strain values for when the hybridization likely occurs? This data is in principle shown in Fig. 3 c for one defect, but a statistic would be nice. For example a scatter plot with strain on one axis, energy of the PL on the other, and peak intensity as color.

-> Is the relative height of the alignment mark in Fig. 2 b,c to the SPE peak relevant?

Minor points

-> Fig. 3 "The shown region" -> "The region shown"

Point-by-point response to the Reviewers' comments

We appreciate all comments and suggestions from the referees. Below we provide our point-to-point response (shown in black) to the original referees' comments (shown in blue). We have revised the manuscript and Supplementary Information files accordingly (the changes are highlighted in red in the manuscript and SI pdf documents).

Reviewer #1 (Remarks to the Author):

The manuscript by A. N. Abramov and colleagues addresses a timely topic concerning the physical origin of single photon emitters in two-dimensional materials. Many groups worldwide study this kind of sources aiming at their exploitation for quantum technologies.

This manuscript presents a series of cleverly conducted experiments. The paper is clearly written and the results well presented. Based on the reported experiments and related analyses the authors confirm and strengthen a previously published hypothesis concerning the mechanism leading to the generation of single photons in 2D materials (Refs 23 and 45). The concomitant presence of moderate strain and of defects triggers the optical emission from otherwise dark excitons. To this respect, the present work does not bring great elements of novelty.

We thank the Reviewer for his/her consideration and generally positive evaluation of our work. We are also grateful for the Reviewer's valuable comments, which allowed us to further improve the quality of our manuscript. We provide our response to the comments below.

In the following, a few technical concerns are listed.

1) Are the emission spectra reproducible in terms of intensity and emitted photon wavelength after each thermal cycle?

The PL spectra measured on our experimental samples are generally reproducible within 2-3 thermal cycles. After that, we start to observe some degradation of the SPE brightness accompanied with nm-scale shifts of the emission wavelength. We believe that the slow changes of the emission characteristics are due to laser-induced heating of the locally strained regions, since the effect is usually more pronounced for measurements with higher excitation power.

2) The relationship between saturation power and emission rate might be not known to all readers. Please, provide a brief explanation; the Supporting Information could be an appropriate place.

Following the Reviewer's suggestion, we have expanded Supplementary Note 4 to clarify the relationship between saturation power and emission rate for the readers. Briefly, the SPE emission rate I exhibits a saturating dependence on pump power P : $I = I_0/(1 + P_{\text{sat}}/P)$, where saturation power P_{sat} is a constant. At high powers, the SPE emission rate approaches the asymptotic value I_0 limited by the SPE quantum yield. We obtain I_0 experimentally as the asymptotic value in the measured power-dependent emission rate curves (Fig. S4a). When we pump close to the saturation power $P \sim P_{\text{sat}}$, the SPE emission rate is approximately half of its maximum achievable value: $I \sim I_0/2$.

3) If feasible, the authors should provide a statistical analysis regarding of characteristics of the observed single photon emitters. In addition to the emission wavelength histogram in Fig. 4 (c), it'd be relevant to present a similar analysis on the single photon emitter linewidth and $g_2(0)$, if available.

In response to the Reviewer's suggestion, we have added more results on the statistics of the studied single photon emitters to the revised Supplementary Notes 5 and 4. In addition to the emission wavelength histogram, Fig. S5 presents results on the measured PL lifetime (d), linewidth as FWHM

(f), and $g_2(0)$ values (b). In order to reliably determine the lifetime, linewidth, and $g_2(0)$, we select only SPEs that exhibit well-resolved individual PL peaks with reasonably high brightness. Therefore, the statistical sample size for these data are smaller than that for the wavelength data presented in Fig. 4c of the main text. Additionally, in the revised Fig. S4 we show statistical data on estimated SPE brightness (b) and quantum yield (c).

4) The matching between the AFM and the optical microscopy measurements is made from images recorded at different temperatures. This is of course fine. Nevertheless, can the authors comment on the reliability of this method? May issues related to the thermal expansion/contraction of the sample be present?

We thank the Reviewer for highlighting this point, which was not mentioned in our original manuscript. Indeed, the AFM and PL measurements are performed at different temperatures. However, considering the relatively low thermal expansion coefficients of both SiO_2 and WSe_2 ($\sim 10^{-6}$ 1/K), the alignment mark grid size of 20 μm , similar size of the monolayer, and the fact that the PMMA layer is in contact with both SiO_2 and WSe_2 , we estimate possible distortions of the linear dimensions due to thermal expansion to be below 10 nm and well within our experimental uncertainty. We check this by comparing optical images of the sample at 10 K and 300 K, where the relative coordinates of the alignment marks and edges/cracks in the monolayer sample do not change within the experimental uncertainty. We note that the error introduced by thermal expansion could become important if one used a substrate with high thermal expansion coefficient or significantly larger alignment grid cell ($> 50 \mu\text{m}$). We have added a statement about the possible thermal expansion issues to the revised manuscript on page 7.

5) Frankly, I could not follow completely the procedure employed to extract the beautiful maps shown in Figures 3 (b) and (c) and in Figures 4 (d) and (e). In particular, I refer to equations (1) and (2) and to the method employed to derive the strain tensor over the interested area. The known quantities are the material's elastic parameters and the AFM height profile (namely, the displacement h). I suppose then that from the vertical displacement, via the material's elastic parameters, the authors retrieve the in-plane displacement vector (u). It looks like a reverse Poisson's effect: known the vertical displacement, we derive the in-plane compression/expansion of the layer. I believe that this important aspect would deserve a more detailed discussion in the Supporting Information.

To clarify the calculation procedure, in the revised manuscript we have extended the explanation of the strain extraction technique. Briefly, the vertical and in-plane displacements are connected, indeed. However, unlike the Poisson's effect, this connection can not be formulated in terms of an analytical expression. Instead, the in-plane displacement field can be derived from the assumption that the 2D crystal is in equilibrium. This requirement can be formalized by the equilibrium equation for the stress tensor, $\sigma \cdot \nabla = 0$. The latter is connected with strain ϵ by Hooke's law (Eq. (2) in the manuscript). Thus, expressing the strain tensor components via the local displacements, we obtain two differential equations which connect the height profile with the in-plane displacement field (u_x, u_y). Solving it with the finite element method, we obtain the map of (u_x, u_y) and restore the full strain tensor using Eq. (1). We have added an extended discussion of the strain extraction procedure in Supplementary Note 7.

6) The discussion at page 9 about the fine-structure splitting is a bit ill-posed given the relatively broad linewidth of the emitter shown in Figure 1 for example.

Indeed, the estimated fine-structure splitting is similar to the linewidth of the studied emitters. This in itself can obscure the observation of the splitting in our experiment. However, if our SPE PL peak consisted of two orthogonally polarized split peaks, we would expect to observe change in the shape or spectral position of the peak for different polarization directions in the detection channel. We do not see such change in polarization-resolved measurements. Therefore we believe that at least part of the reason for measuring only a single SPE PL peak is related to the predominant population of the lowest state in the fine-structure doublet at low temperature. We have revised the discussion on page 9 to clarify that the observation of peak splitting is also obscured due to the relatively broad linewidth.

7) Figure 4 (c) is very suggestive. I wonder if the vertical axis of the left panel corresponds to a true energy scale. Namely, do the intervals spanned by the horizontal lines relative to X_{bright} and X_{dark} correspond to a specific strain value?

A curiosity: what do the authors expect to observe (or have observed) if X_{bright} is resonant with the defect level? Actually, this seems to be a real possibility given the large tensile strain values present in the imprinted monolayers (dark blue regions in Figures 3 (b) and (c) and in Figures 4 (d) and (e)).

The sequence of the horizontal lines shown in the left panel of Fig. 4c in the original manuscript simply illustrates the effect of the applied strain and does not represent actual experimental data. To avoid confusion, we have revised the left panel of Fig. 4c to appear as a schematic illustration.

Regarding the second part of the comment, we expect that tuning X_{bright} into resonance with the vacancy level would not lead to the appearance of any significant SPE emission. Since the strain-induced energy shift of the bright and dark excitons are similar, the dark exciton state remains lower in energy for all considered strain values. Therefore, the exciton population would be still predominantly in the dark state. These estimations are partially confirmed by the measurements published in [P. H. Lopez et al., Nat. Commun. 13 7691 (2022)] where no manifestations of the hybridization between bright exciton and Se-vacancies was detected in a wide range of the applied strains.

In summary, this is a well written manuscript presenting a series of interesting experimental results and elegant numerical analysis (provided this latter is more clearly presented and detailed). The main conclusions of this work match with those previously reported and therefore the novelty of this work is somehow undermined. My opinion is that npj 2D materials and applications could be a more suitable journal with the advantage of a rapid transfer and publication.

We thank the Reviewer again for the generally positive assessment of our manuscript. Following the Reviewer's suggestions, we have further improved the manuscript in several aspects, including a more detailed presentation of the numerical analysis idea, procedure, and results. Regarding the Reviewer's comment on novelty, we believe that there are two aspects where our work provides significant advance over previously published results. First, it establishes a general method for probing the effect of complex local nanoscale strain distributions in a 2D membrane on the properties of individual quantum emitters, which includes a straightforward recipe for modeling strain profiles based on the experimentally known surface topography data. Second, our results provide an independent and rather direct way for verifying the underlying mechanism of single-photon emission. While the defect-related mechanism of SPE formation in WSe₂ monolayer discussed in our manuscript has been proposed previously, our results confirm that it is indeed the most likely mechanism and therefore help resolve the important existing debate in the literature, as also highlighted by Reviewer 2 in their report.

Reviewer #2 (Remarks to the Author):

The manuscript by Abramov et al. describes their study of strain induced single photon emitters in monolayer WSe₂ using a nanoindentation process. Through PL imaging and atomic force microscopy, they are then able to deduce the amount of strain at the vicinity of the emitter. This then allows them to calculate what the strain induced shift of the dark exciton in WSe₂ would be at the emitter location. From here, they show a high degree of correlation between the energy shifted dark exciton wavelength and the emitter wavelength.

While neither the nanoindentation process for generating single photon emitters nor the PL/AFM imaging for locating a quantum emitter are novel, I do feel that the overall conclusions of the paper are very noteworthy. The underlying mechanisms for strain induced single photon emission in monolayer WSe₂ has not been well understood, with one prominent theory being exciton funneling and band shifting at the strain locations. The work presented here shows that exciton funneling is not correct and that hybridization of dark exciton states with Se defects is the most probable cause. These results will

be of much interest to those working in quantum emission from 2D materials and, as such, I feel the manuscript is worthy of consideration for publication in Nature Communications.

We thank the Reviewer for appreciating our work and providing valuable comments that helped us further improve the quality of our manuscript.

Below are a few technical comments:

1) In figure 1d, the fit definitely extends well below the histogram data point at $g_2(0)$, with the data point appearing to be roughly $g_2(0) = 0.1$. This fact, coupled with the level of noise in the $g_2(t)$ spectra, makes it hard for me to be fully convinced that the $g_2(0)$ value is 0.02 as claimed. Calculating and listing the uncertainty in the fit might be helpful.

We agree with the Reviewer's comment that it is important to take into account not only the values of $g_2(0)$ obtained from fitting but also the associated uncertainties. The uncertainty of the fit shown in Fig. 1d is indeed as large as 0.13 due to the noise present in the data. Therefore in the manuscript we mention the result $g_2(0) = 0.02$ only as a single value extracted from the fit and not the characteristic value for the studied emitters. However, our statistics shown in Fig. S4 of the revised Supplementary Information demonstrates that for a significant portion of the measured SPEs fitting gives $g_2(0)$ values below 0.1. The overall average and standard deviation obtained from the statistics of measured correlation functions is $g_2(0) = 0.15 \pm 0.12$. To clarify this point, we have revised the corresponding statement in the manuscript on page 5.

2) In supplementary note 3, I think it might be illustrative to include an additional figure showing again the filtered spectra but this time with it normalized to the unfiltered SPE PL. Maybe it is just due to the normalization but to me it appears that you are able to reduce the background signal by more than 50% using polarization filtering which I do not believe should be possible. To my understanding, this background signal is usually unpolarized.

The Reviewer is absolutely right that the background signal is unpolarized. We have double checked the original figure and found out that, in addition to the different normalization factors, it contained spectra taken at slightly different conditions. We apologize for the confusion and thank the Reviewer for pointing out this issue. We have revised Fig. S3 and Supplementary Note 3 to show PL spectra taken at the same conditions in co- and cross-polarization channels. Additionally, in the insets of the plots we show the full dependence of the PL spectrum on the polarization direction, which demonstrates a two-lobed pattern for the SPE and uniform pattern for the background.

3) In supplementary note 5, the text references figure S3 when it should be figure S5. Please check the manuscript for other such small typos.

We have corrected this typo and carefully checked the entire manuscript for other possible typos.

Reviewer #3 (Remarks to the Author):

Abramov and co-authors present measurements on single photon emitters in strained WSe₂. They provide spatially resolved AFM images aligned with optical PL images, to quite accurately pinpoint the location of the single photon emitters relative to local strain.

Single photon emitters are a key ingredient of many quantum technologies. The present paper confirms strain-induced hybridization between conduction band states and localized defect states as their origin in WSe₂. The paper is well written, the topic at hand is timely, and the contribution significant. I therefore recommend publication in Nature Communication.

We thank the Reviewer for his/her consideration and positive evaluation of our work. We also thank the Reviewer for providing valuable suggestions that helped us further improve the manuscript.

Suggestions:

-> As far as I understand, the method by the authors would allow for changing the relative position between the strain-inducing afm tip and the defect. Have the authors tried this, or is the number of defects too large and the membrane relaxation too big to find the same defect again?

In principle, our method allows precise positioning of the tip with respect to a single defect after the coordinates of the defect are determined. However, we know the coordinates of the defect only after we locally strain the monolayer via deforming the polymer substrate in an irreversible way. In experiments, we observe that further attempts to create locally strained regions within 1 μm distance of the original indent result in the disappearance of single-photon emission from existing SPEs. This agrees with the expectations from our theoretical model, since any further deformations in the vicinity of the original strained region will likely change the local strain tensor components at the defect location and bring the dark exciton out of resonance with the defect energy level.

-> Can the authors provide quantitative strain values for when the hybridization likely occurs? This data is in principle shown in Fig. 3 c for one defect, but a statistic would be nice. For example a scatter plot with strain on one axis, energy of the PL on the other, and peak intensity as color.

In response to the Reviewer's suggestion, we have added the requested analysis in the Supplementary Note 8. The scatter plot contains the data for the total of 9 emitters for which the position, linewidth and the emission intensity have been simultaneously measured. The obtained results demonstrate a good coincidence with the strain-induced hybridization interpretation of the SPE origin in monolayer WSe_2 .

-> Is the relative height of the alignment mark in Fig. 2 b,c to the SPE peak relevant?

In our experiment, we control the signal from the alignment marks by changing the intensity of the illumination by the lamp. On the one hand, we aim to minimize the uncertainty for the measured alignment mark coordinates via increasing the illumination intensity. On the other hand, this simultaneously increases the intensity of parasitic reflection from the substrate, which reduces the signal-to-noise ratio for SPE PL and leads to an increased uncertainty for the measured SPE coordinates. Therefore, we choose the optimal intensity of illumination in such a way to achieve similar uncertainties for the measured alignment mark and SPE coordinates, which results in the minimized error for the finally calculated SPE position. These considerations dictate the relative height of the peaks shown in Fig. 2 b,c. We have added this clarification in the revised manuscript on pages 5-6.

Minor points

-> Fig. 3 "The shown region" -> "The region shown"

We have corrected the phrasing.

REVIEWERS' COMMENTS

Reviewer #1 (Remarks to the Author):

The authors addressed my (as well as the other reviewers') previous concerns in a satisfactory manner. Apart from its technical and scientific merit, I am now convinced that the present manuscript is of broad interest and general validity to be published in Nature Commun. As a matter of fact, very recently it was reported (Adv. Optical Mater. 11, 2202953 (2023)) the observation of single photon emitters in strained WS₂ monolayers (another "darkish" TMD), where the emitters are not located in the regions of maximum strain but at the edges of the deformed structures like in the present work.

Reviewer #2 (Remarks to the Author):

The revised version of the manuscript has adequately addressed all of my concerns and I recommend it for publication in Nature Communications.

Reviewer #3 (Remarks to the Author):

The authors have adequately addressed my comments - from my side the paper can now be published in Nature Comm.. Concerning the point of novelty of the manuscript, I find the response of the authors convincing.

Three minor points the authors should address:

-> I find the point raised by the authors in reply to question 1) of referee 1 (reproducibility of spectra) important: slow changes of emission characteristics after a few cycles due to laser-induced heating. I would suggest to mention this somewhere in the revised supplement.

-> Likewise, I would suggest discussing the arguments related to my original question concerning repositioning the AFM tip with respect to the defect somewhere in the supplement.

-> Reference 41 (Lopez et al.) still is referred to by arXiv identifier in the main manuscript, the paper has meanwhile appeared as Nature Comm. 13, 7691 (2022).

Point-by-point response to the Reviewers' comments

We thank all Reviewers for their feedback. Below we provide our response (shown in black) to the additional comments of the Reviewers (shown in blue).

Reviewer #1 (Remarks to the Author):

The authors addressed my (as well as the other reviewers') previous concerns in a satisfactory manner. Apart from its technical and scientific merit, I am now convinced that the present manuscript is of broad interest and general validity to be published in Nature Commun. As a matter of fact, very recently it was reported (Adv. Optical Mater. 11, 2202953 (2023)) the observation of single photon emitters in strained WS₂ monolayers (another "darkish" TMD), where the emitters are not located in the regions of maximum strain but at the edges of the deformed structures like in the present work.

We thank the Reviewer for considering our work and recommending it for publication. We have added a reference to the relevant work suggested by the Reviewer to the final version of our manuscript.

Reviewer #2 (Remarks to the Author):

The revised version of the manuscript has adequately addressed all of my concerns and I recommend it for publication in Nature Communications.

We thank the Reviewer again for considering our work and recommending it for publication.

Reviewer #3 (Remarks to the Author):

The authors have adequately addressed my comments - from my side the paper can now be published in Nature Comm.. Concerning the point of novelty of the manuscript, I find the response of the authors convincing.

We thank the Reviewer for considering our work and recommending it for publication.

Three minor points the authors should address:

-> I find the point raised by the authors in reply to question 1) of referee 1 (reproducibility of spectra) important: slow changes of emission characteristics after a few cycles due to laser-induced heating. I would suggest to mention this somewhere in the revised supplement.

We have added a corresponding statement to the revised Supplementary Note 4 on page S6.

-> Likewise, I would suggest discussing the arguments related to my original question concerning repositioning the AFM tip with respect to the defect somewhere in the supplement.

We have added a corresponding discussion to the revised Supplementary Note 1 on page S2.

-> Reference 41 (Lopez et al.) still is referred to by arXiv identifier in the main manuscript, the paper has meanwhile appeared as Nature Comm. 13, 7691 (2022).

We have updated the reference accordingly.